# Controllable Unsupervised Text Attribute Transfer via Editing Entangled Latent Representation

**Ke Wang**     **Hang Hua**     **Xiaojun Wan**
Wangxuan Institute of Computer Technology, Peking University
The MOE Key Laboratory of Computational Linguistics, Peking University
`{wangke17, huahang, wanxiaojun}@pku.edu.cn`

## Abstract

Unsupervised text attribute transfer automatically transforms a text to alter a specific attribute (e.g. sentiment) without using any parallel data, while simultaneously preserving its attribute-independent content. The dominant approaches are trying to model the content-independent attribute separately, e.g., learning different attributes' representations or using multiple attribute-specific decoders. However, it may lead to inflexibility from the perspective of controlling the degree of transfer or transferring over multiple aspects at the same time. To address the above problems, we propose a more flexible unsupervised text attribute transfer framework which replaces the process of modeling attribute with minimal editing of latent representations based on an attribute classifier. Specifically, we first propose a Transformer-based autoencoder to learn an entangled latent representation for a discrete text, then we transform the attribute transfer task to an optimization problem and propose the Fast-Gradient-Iterative-Modification algorithm to edit the latent representation until conforming to the target attribute. Extensive experimental results demonstrate that our model achieves very competitive performance on three public data sets. Furthermore, we also show that our model can not only control the degree of transfer freely but also allow transferring over multiple aspects at the same time.[1]

## 1   Introduction

Text attribute transfer is a task of editing a text to alter specific attributes, such as sentiment, style and tense [17]. It has drawn much attention in the natural language generation field and it is a requirement of a controllable natural language generation system. Given a source text with an attribute (e.g., positive sentiment), the goal of the task is to generate a new text with a different attribute (e.g., negative sentiment). The generated text should meet the requirements: $(i)$ maintaining the attribute-independent content as the source text, $(ii)$ conforming to the target attribute and $(iii)$ still maintaining the linguistic fluency. However, due to the lack of parallel corpora exemplifying the desired transformations between source and target attributes, most approaches [5, 10, 28, 26, 29, 20, 40, 41, 38, 17, 16] are unsupervised and can only access non-parallel or monolingual data.

The dominant methods of unsupervised text attribute transfer are to separately model attribute and content representations, such as using multiple attribute-specific decoders [5] or combining the content representations with different attribute representations to decode texts with target attribute in an adversarial [10, 28, 26, 29, 20, 40, 41, 38, 17] or non-adversarial [16] way. Nevertheless, such practices have shortcomings. First, because they try to disentangle attribute and attribute-independent content, this may undermine the integrity (i.e., naturality) and result in poor readability of the

generated sentences. Second, they require modeling each new attribute separately and thus lack flexibility and controllability.

To address the above problems, we propose a controllable unsupervised text attribute transfer framework, which not only can flexibly control the degree of transfer, but also can control transfer over multiple aspects (e.g., modification of sentiments towards multiple aspects in a text) at the same time. We achieve this goal by modifying the source text's latent representation. Different from the mainstream methods [10, 28, 26, 29, 20, 40, 41, 38], which learn the attribute and content representations separately and then decode the text with the target attribute, our latent representation is an entangled representation of both attribute and content. Our model consists of a Transformer-based autoencoder and an attribute classifier. We first train the autoencoder and the classifier separately and use the encoder to get the latent representation of the source text, and then we use our proposed Fast-Gradient-Iterative-Modification (FGIM) algorithm to iteratively edit the latent representation, until the latent representation can be identified as target attribute by the classifier. So that the target text can be decoded from the modified latent representation by the decoder.

Our contributions are summarized as follows: (1) We build a Transformer-based autoencoder with low reconstruction bias to learn an entangled latent representation for both attribute and content, rather than treating them separately, so the integrity of the language expressions will not be damaged. And the decoded target text will keep natural and fluent (requirement $iii$). (2) We design a Fast-Gradient-Iterative-Modification (FGIM) algorithm by using a well-trained attribute classifier to provide an appropriate modification direction for the latent representation, so we will modify the latent representation as little as possible (requirement $i$) until conforming to the target attribute (requirement $ii$). (3) Our method is capable of controlling text attribute in a more flexible way, e.g., controlling the degree of attribute transfer and allowing to transfer over multiple aspects. (4) Our method achieves very competitive performance on three datasets, especially in terms of text fluency and transfer success rate.

## 2 Related Work

**Text Attribute Transfer:** Text attribute transfer [39, 22, 11] is a type of conditional text generation [14, 31, 1, 4, 10, 37, 20, 36], inspired by visual style transfer [12, 7, 44, 18]. Recently, various approaches have been proposed for handling textual data, mainly aiming at con the writing style of sentences. However, it is hard to find large scale datasets of parallel sentences written in different styles [22]. Li et al. [17] released three small crowd-sourced text style transfer datasets for evaluation purposes, where the sentiment had been swapped (between positive and negative) while preserving the content. Controlled text attribute transfer from unsupervised data is thus the focus of recent researches.

In general, related researches on unsupervised text attribute transfer are divided into two main categories, phrase based and latent representation based. The phrase-based approaches [17, 38] explicitly separate attribute phrases from attribute-independent content phrases and replace them with phrases of target attribute. But it may damage the overall consistency of the sentence and make the generated text unnatural [33]. Another is to learn latent representations, and most solutions use adversarial methods [10, 28, 26, 5, 29, 20, 40, 41] to learn the latent representation of attribute and content, and then pass through different attribute-specific decoders or combining attribute and content latent representations to generate a variation of the input sentence with different attribute.

As mentioned above most approaches learn the attribute and content representations separately. However, the generated text of such methods may have poor readability. Besides, Lample et al. [16]'s experiment also shows there is no need to disentangle the attribute and content. Lample et al. [16]'s work is most related to ours, but in Lample et al. [16]'s approach, it still needs an extra attribute embedding to control the attribute of generated text. In this study, we try to explore an unsupervised attribute transfer method that only needs to iteratively edit the entangled latent representation of attribute and content.

**Adversarial Samples Generation:** Our work is also related to adversarial samples generation [8, 42], which also uses the adversarial gradient to edit continuous samples to change classifier's predictions. However, different from them, we edit on the latent space and then decode samples, rather than directly editing the samples. In addition, we want to generate meaningful samples that match the goals, rather than producing a small perturbation to fool the classifier.

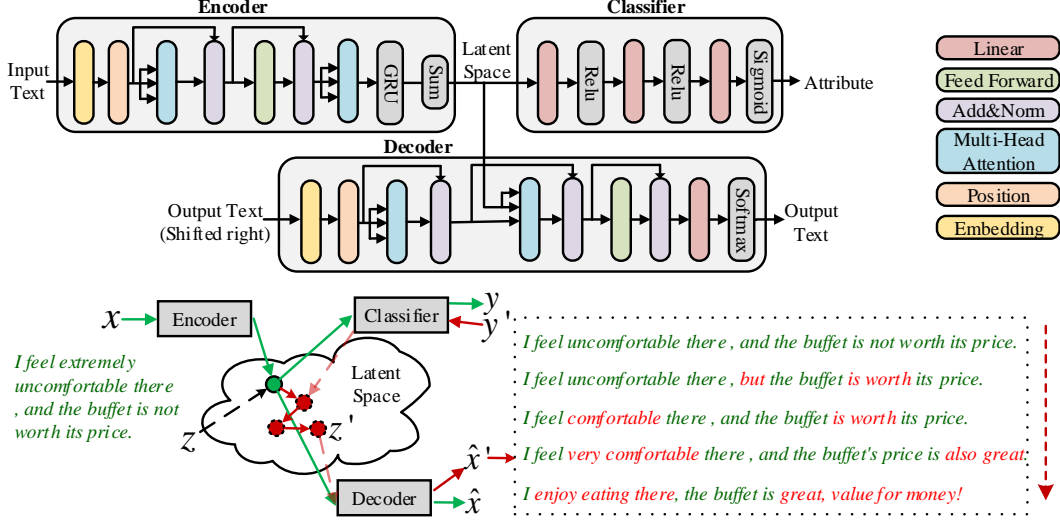

Figure 1: Model architecture.

**Activation Maximization:** Our work is also related to the activation maximization methods [3, 24, 23, 43], which synthesize an input (e.g. an image) that highly activates a neuron. Although these methods have been successfully applied to producing generative models, they are generally limited to continuous spaces (e.g., images, etc.) because the gradients are difficult to pass in discrete spaces. Different from these studies [24, 23, 43], our method is the first to apply activation maximization to text generation, which consists of two aspects: 1) encoding discrete texts into contiguous latent spaces with an autoencoder. 2) modifying the latent representations based on the direction that highly actives the classifier.

## 3 Model

### 3.1 Problem Formalization

We consider a dataset $\boldsymbol{X}$, which has $n$ sentences, and each sentence $\boldsymbol{x}$ is paired with an attribute vector $\boldsymbol{y}$. For example, we can use $\boldsymbol{y} = (y_{tense}, y_{sentiment})$ to represent both "tense" and "sentiment" attributes of a sentence, or use $\boldsymbol{y} = (y_{appearance}, y_{aroma}, y_{palate}, y_{taste}, y_{overall})$ to represent the sentiment types or values on five aspects of a beer review. In most cases, $\boldsymbol{y}$ actually contains only one attribute, e.g., the overall sentiment. In general, given source text $\boldsymbol{x}$ and target attribute $\boldsymbol{y'}$, text attribute transfer seeks to generate fluent target text $\hat{\boldsymbol{x}}'$, which preserves the original attribute-independent content but conforms to the target attribute $\boldsymbol{y'}$.

### 3.2 Model Overview

The architecture of our proposed model is depicted in Figure 1. The whole framework can be divided into three sub-models: an encoder $E_{\theta_e}$ which encodes the text $\boldsymbol{x}$ into a latent representation $\boldsymbol{z}$, a decoder $D_{\theta_d}$ which decodes text $\hat{\boldsymbol{x}}$ from $\boldsymbol{z}$, and an attribute classifier $C_{\theta_c}$ that classifies attribute of the latent representation $\boldsymbol{z}$. That is:

$$\boldsymbol{z} = E_{\theta_e}(\boldsymbol{x}); \ \boldsymbol{y} = C_{\theta_c}(\boldsymbol{z}); \ \hat{\boldsymbol{x}} = D_{\theta_d}(\boldsymbol{z}). \tag{1}$$

Formally, in this work, we formulate the text attribute transfer task as an optimization problem. More specifically, we first propose a Transformer-based autoencoder to learn a latent representation $\boldsymbol{z} = E_\theta(\boldsymbol{x})$ of a discrete text, which is entangled with content and attribute. Then, the task of finding the target text $\hat{\boldsymbol{x}}'$ with target attribute $\boldsymbol{y'}$ can be formulated as the following optimization problem:

$$\hat{\boldsymbol{x}}' = D_{\theta_d}(\boldsymbol{z}') \ where \ \boldsymbol{z}' = argmin_{\boldsymbol{z}^*}||\boldsymbol{z}^* - E_{\theta_e}(\boldsymbol{x})|| \ s.t. \ C_{\theta_c}(\boldsymbol{z}^*) = \boldsymbol{y'}. \tag{2}$$

To solve this problem, we propose the Fast-Gradient-Iterative-Modification algorithm (FGIM), which modifies $z$ based on the gradient of back-propagation by linearizing the attribute classifier's loss function on $z$.

In brief, we transform the original problem to find an optimal representation $z'$ that conforms to the target attribute $y'$ (requirement $ii$) and is "closest" to $z$ (requirement $i$), then we decode the target text $\hat{x}'$ from $z'$ (requirement $iii$).

**Transformer-based Autoencoder:** One of the key points of our model is to build an autoencoder with low reconstruction bias. Inspired by the superiority of Transformer [34] on many text generation tasks [34, 27, 2], we propose a Transformer-based autoencoder with low reconstruction bias to learn the latent representation of source text. We first pass source text $x$ through the original Transformer's encoder ($E_{transformer}$) [34] and get the intermediate representations $U$. Because the Transformer architecture is suboptimal for language modelling itself, neither self-attention nor positional embedding in the Transformer is able to effectively incorporate the word-level sequential context [35]. So we add extra positional embeddings $H$ [34] to $U$. Next we pass $U$ through a GRU layer with self-attention to further utilize the sequence information. Then we apply a sigmoid activation function on the GRU hidden representations and sum them to get the final latent representation $z$ (Figure 1):

$$z = E_{\theta_e}(x) = Sum(Sigmoid(GRU(U + H))), where\ U = E_{transformer}(x). \qquad (3)$$

Finally the target text $\hat{x}$ can be decoded from $z$. During the autoencoder optimization process, we adopt the label smoothing regularization [32] to improve the performance of the model. Hence, our autoencoder reconstruction loss is:

$$\mathcal{L}_{ae}(D_{\theta_d}(E_{\theta_e}(x)), x) = \mathcal{L}_{ae}(D_{\theta_d}(z), x) = -\sum^{|x|}((1 - \varepsilon)\sum_{i=1}^{v}\bar{p}_i\log(p_i) + \frac{\varepsilon}{v}\sum_{i=1}^{v}\log(p_i)), \quad (4)$$

where $v$ denotes the vocabulary size, and $\varepsilon$ denotes the smoothing parameter. The last item ($\frac{\varepsilon}{v}\sum_{i=1}^{v}\log(p_i)$) is the introduction of noise to relax our confidence in the label. For each time step, $p$ and $\bar{p}$ are the predicted probability distribution and the ground truth probability distribution over the vocabulary, respectively.

**Attribute Classifier for Latent Representation:** In our framework, we use an attribute classifier to provide the direction (gradient) for editing the latent representation so that it conforms to the target attribute. Our classifier is two stacks of linear layer with sigmoid activation function, and the attribute classification loss is:

$$\mathcal{L}_c(C_{\theta_c}(z), y) = -\sum_{i=1}^{|q|}\bar{q}_i\log q_i, \qquad (5)$$

where $q$ represents the predicted attribute probability distribution and $\bar{q}$ is the true attribute probability distribution. Additionally, in practice we find it benefits the results to optimize the above two loss functions separately, rather than training them jointly.

### 3.3 Fast Gradient Iterative Modification Algorithm

The goal of editing the latent representation is to transfer from the source attribute to the target attribute. That is to find an optimal representation $z'$, which is "closest" to $z$ in the latent space and conforms to the target attribute $y'$. Inspired by Goodfellow et al. [8], we ascertain the fastest modification direction with the gradient back-propagation of attribute classification loss calculation. More specifically, to get an optimal $z'$, we first use $z$ as the input of $C_{\theta_c}$ and use $y'$ as the label to calculate the gradient to $z$. Then we modify $z$ in this direction iteratively until we get a $z'$ that can be identified as the target attribute $y'$ by the classifier $C_{\theta_c}$. Note that the gradient is computed with respect to the input $z$, instead of the model parameters $\theta_c$. In other words, we use the gradient to change $z$ rather than change model parameters $\theta_c$. In each iteration, the newly modified latent representation $z^*$ can be formulated as:

$$z^* = z - w_i\nabla_z\mathcal{L}_c(C_{\theta_c}(z), y'), \qquad (6)$$

where $w_i$ is the modification weight used for controlling the degree of transfer. Contrary to Goodfellow et al. [8], we want a modification to make the latent representation more different in attribute, but not

a tiny adversarial perturbation to fool the classifier. Thus we propose a Dynamic-weight-initialization method to allocate the initial modification weight $w_i$ in each trial process. More specifically, we give a set of weights $\boldsymbol{w} = \{w_i\}$, and our algorithm will dynamically try each weight in $\boldsymbol{w}$ from small to large until we get our target latent representation $\boldsymbol{z}'$. This will prevent the modification of $\boldsymbol{z}$ from falling into local optimum. In each trial process, the initial weight $w_i \in \boldsymbol{w}$ will iteratively decay by multiplying a fixed decay coefficient $\lambda$. Our algorithm is shown in Alg 1.

---

**Algorithm 1** Fast Gradient Iterative Modification Algorithm.

---

**Input:** Original latent representation $\boldsymbol{z}$; Well-trained attribute classifier $C_{\theta_c}$; A set of weights $\boldsymbol{w} = \{w_i\}$;
    Decay coefficient $\lambda$; Target attribute $\boldsymbol{y}'$; Threshold $t$;
**Output:** An optimal modified latent representation $\boldsymbol{z}'$;

 1: **for** each $w_i \in \boldsymbol{w}$ **do**
 2:     $\boldsymbol{z}^* = \boldsymbol{z} - w_i \nabla_{\boldsymbol{z}} \mathcal{L}_c(C_{\theta_c}(\boldsymbol{z}), \boldsymbol{y}')$;
 3:     **for** s-steps **do**
 4:         **if** $|\boldsymbol{y}' - C_{\theta_c}(\boldsymbol{z}^*)| < t$ **then** $\boldsymbol{z}' = \boldsymbol{z}^*$ ; Break;
 5:         **end if**
 6:         $w_i = \lambda w_i$;
 7:         $\boldsymbol{z}^* = \boldsymbol{z}^* - w_i \nabla_{\boldsymbol{z}^*} \mathcal{L}_c(C_{\theta_c}(\boldsymbol{z}^*), \boldsymbol{y}')$;
 8:     **end for**
 9: **end for**
10: **return** $\boldsymbol{z}'$;

---

Our **Fast Gradient Iterative Modification** algorithm has the following advantages:

**Attribute Transfer over Multiple Aspects:** Compared to the methods which use extra attribute embedding [28, 10] or multi-decoder [5], our proposed framework transfers the source text's attribute into any target attribute by using only the classifier $C_{\theta_c}$ and the target attribute $\boldsymbol{y}$. One of the advantages of our model is the flexibility to design the goals of the attribute classifier to achieve the attribute transfer over multiple aspects, which no other models have attempted.

**Transfer Degree Control:** Our model can use different modification weight in $\boldsymbol{w}$ to control the degree of modification, thus achieving the control of the degree of attribute transfer, which is never considered by other models.

## 4 Experiment

### 4.1 Implementation

In our Transformer-based autoencoder, the embedding size, the latent size and the dimension size of self-attention are all set to 256. The hidden size of GRU and batch-size are set to 128. The inner dimension of Feed-Forward Networks (FFN) in Transformer is set to 1024. Besides, each of the encoder and decoder is stacked by two layers of Transformer. The smoothing parameter $\varepsilon$ is set to 0.1. For the classifier, the dimensions of the two linear layers are 100 and 50. For our FGIM, the weight set $\boldsymbol{w}$, the threshold $t$ and the decay coefficient $\lambda$ are set to $\{1.0, 2.0, 3.0, 4.0, 5.0, 6.0\}$, 0.001 and 0.9, respectively. The optimizer we use is Adam [15] and the initial learning rate is 0.001. We implement our model based on Pytorch 0.4.

### 4.2 Datasets

We use datasets provided in Li et al. [17] for sentiment and style transfer experiments, where the test sets contain human-written references.

**Yelp:** This dataset consists of Yelp reviews for flipping sentiment. We consider reviews with a rating above three as positive samples and those below three as negative ones;

**Amazon:** This dataset consists of product reviews from Amazon [9] for flipping sentiment. Similar to Yelp, we label the reviews with a rating higher than three as positive and less than three as negative;

**Captions:** This dataset consists of image captions [6] for changing between romantic and humorous styles. Each caption is labeled as either romantic or humorous.

It is worth noting that there are only manual reference answers on the test set. The statistics of the above three datasets are shown in Table 1.

Table 1: Statistics for Yelp, Amazon, Captions datasets.

| Dataset | Styles | #Train | #Dev | #Test | #Vocab | Max-Length | Mean-Length |
|---------|--------|--------|------|-------|--------|------------|-------------|
| Yelp | Negative | 180,000 | 2,000 | 500 | 9,640 | 15 | 8.89 |
| | Positive | 270,000 | 2,000 | 500 | | | |
| Amazon | Negative | 277,000 | 1,015 | 500 | 58,991 | 34 | 14.84 |
| | Positive | 278,000 | 985 | 500 | | | |
| Captions | Humorous | 6,000 | 300 | 300 | 8,693 | 20 | 14.04 |
| | Romantic | 6,000 | 300 | 300 | | | |

## 4.3 Sentiment and Style Transfer Results

We compare our model with eight state-of-the-art models, including CrossAlign [28], MultiDec [5], StyleEmb [5], CycleRL [38], BackTrans [26], RuleBase [17], DelRetrGen [17] and UnsupMT [41].

**Automatic Evaluation:** Following previous works [28, 17, 41], we evaluate models' performance from three aspects: 1) Acc: we measure the attribute transfer accuracy of the generated texts with a fastText classifier [13] trained on the training data; 2) BLEU [25]: we use the multi-BLEU[2] metric to calculate the similarity between the generated sentences and the references written by human; 3) PPL: we measure the fluency of the generated sentences by the perplexity calculated with the language model trained on the respective corpus. The language model is borrowed from the language modeling toolkit - SRILM [30]. The results are shown in Table 2.

From the results, we can see that: 1) Phrase-based methods (e.g., RuleBase [17]) are not good at keeping fluency, even if they have achieved high BLEU scores. 2) The attribute accuracy and BLEU scores of the sentences generated by our model are promisingly high, indicating that our model can effectively modify attributes without changing too much attribute-independent content. 3) The sentences generated by our model are more fluent than that of baseline models. Overall, our model performs better than all baseline models over all metrics on the Yelp and Amazon datasets, and outperforms most of the baseline models on the Captions datasets.

**Human Evaluation:** Further, we conduct a human evaluation to evaluate the quality of generated sentences more accurately. For each dataset, we randomly extract 200 samples (i.e., 100 sentences generated for each target attribute, e.g., positive → negative and negative → positive, or humorous → romantic and romantic → humorous.) and then hire three workers on Amazon Mechanical Turk (AMT) to score each of the items from three aspects: the attribute accuracy (Att), the retainment of content (Con) and the fluency of sentences (Gra). Moreover, we divided the test samples of each model on each task into the same number of small sets, and the same person annotated the same task for all the models. The scores range from 1 to 5, and 5 is the best. The final average scores are shown in Table 3.

Table 2: Automatic evaluation results. ↓ means the smaller the better. We underline the results of our model and bold the best results.

| Methods | Yelp | | | Amazon | | | Captions | | |
|---------|------|------|------|--------|------|------|----------|------|------|
| | Acc | BLEU | PPL↓ | Acc | BLEU | PPL↓ | Acc | BLEU | PPL↓ |
| CrossAlign [28] | 72.3% | 9.1 | 50.8 | 70.3% | 1.9 | 66.2 | 78.3% | 1.8 | 69.8 |
| MultiDec [5] | 50.2% | 14.5 | 84.5 | 67.3% | 9.1 | 60.3 | 68.3% | 6.6 | 60.2 |
| StyleEmb [5] | 10.2% | 21.1 | 47.9 | 43.6% | 15.1 | 60.1 | 56.2% | 8.8 | 57.1 |
| CycleRL [38] | 53.6% | 18.8 | 98.2 | 52.3% | 14.4 | 183.2 | 45.2% | 5.8 | 50.3 |
| BackTrans [26] | 93.4% | 2.5 | 49.5 | 84.6% | 1.5 | 48.3 | 78.3% | 1.6 | 68.3 |
| RuleBase [17] | 80.3% | 22.6 | 66.6 | 67.8% | 33.6 | 52.1 | 85.3% | **19.2** | 35.6 |
| DelRetrGen [17] | 88.8% | 16.0 | 49.6 | 51.2% | 29.3 | 55.4 | 90.4% | 12.0 | 33.4 |
| UnsupMT [41] | 95.2% | 22.8 | 53.9 | 84.2% | 33.9 | 57.9 | **95.5%** | 12.7 | 31.2 |
| Ours | **95.4**% | **24.6** | **46.2** | **85.3%** | **34.1** | **47.4** | 92.3% | 17.6 | **23.7** |

Table 3: Human evaluation results. The kappa coefficient of the three workers is $0.56 \in (0.41, 0.60)$, which means that the consistency is moderate.

| Methods | Yelp | | | Amazon | | | Captions | | |
|---|---|---|---|---|---|---|---|---|---|
| | Att | Con | Gra | Att | Con | Gra | Att | Con | Gra |
| CrossAlign [28] | 2.5 | 2.8 | 3.3 | 2.7 | 2.7 | 3.1 | 2.1 | 2.5 | 3.0 |
| MultiDec [5] | 2.3 | 3.1 | 2.7 | 2.6 | 2.9 | 2.9 | 2.5 | 2.6 | 2.9 |
| StyleEmb [5] | 2.6 | 3.0 | 2.9 | 3.1 | 2.8 | 3.2 | 2.3 | 3.1 | 3.0 |
| CycleRL [38] | 2.9 | 3.0 | 3.2 | 3.2 | 3.1 | 3.2 | 2.5 | 2.9 | 2.8 |
| BackTrans [26] | 2.0 | 2.4 | 2.9 | 2.6 | 2.8 | 3.4 | 2.4 | 2.8 | 2.8 |
| RuleBase [17] | 3.4 | 3.2 | 3.4 | 3.6 | 3.7 | 3.8 | 2.6 | 3.1 | 3.0 |
| DelRetrGen [17] | 3.2 | 2.9 | 3.0 | 3.7 | 3.6 | 3.4 | 2.5 | 2.9 | 3.2 |
| UnsupMT [41] | 3.2 | 3.3 | 3.5 | 3.7 | 4.0 | 3.7 | 2.8 | 2.8 | 3.3 |
| Ours | **3.6** | **3.5** | **3.8** | **4.0** | **4.2** | **4.1** | **3.5** | **3.4** | **3.5** |

As can be seen from the results, our model outperforms baselines by a wide margin on all metrics, which demonstrates the effectiveness of our proposed Transformer-based autoencoder and FGIM algorithm. Moreover, texts generated by our model have better fluency and attribute accuracy. Interestingly, in our experiments, the results of manual and automatic evaluations are not always consistent with each other, which deserves further study. But even in the eyes of humans, our model excels in the preservation of content, indicating that our model loses less information while achieving the goal of attribute transfer. We show some examples generated by the models in **Supplementary Material**.

## 4.4 Multi-Aspect Sentiment Transfer

In order to evaluate the capability of multi-aspect sentiment transfer of our model, we use a Beer-Advocate dataset, which was scraped from Beer Advocate [19]. Beer-Advocate is a large online review community boasting 1,586,614 reviews of 66,051 distinct items composed by 33,387 users. Each review is accompanied by five numerical ratings over five aspects of "appearance", "aroma", "palate", "taste" and "overall" (here we simply treat "overall" as a special aspect), and each rating is normalized into [0, 1].

As far as we know, there are no previous works investigating aspect-based sentiment transfer, because it is difficult to disentangle sentiment attributes from multiple different aspects or learn so many different combinations of aspect-based sentiments. However, our model can achieve this goal by training the corresponding aspect-based sentiment classifier. We train our Transformer-based autoencoder on this dataset using $\mathcal{L}_{ae}$, and then we train our aspect-based sentiment classifier by the new five-dimension attribute vector $\boldsymbol{y} = \{y_{appearance}, y_{aroma}, y_{palate}, y_{taste}, y_{overall}\}$, which means five sentiment values of a beer review towards five aspects. For evaluation, we randomly sample 300 items to perform the multi-aspect sentiment transfer. For the sake of simplicity, we aim to transform 150 texts into texts with all negative sentiments over five aspects $\boldsymbol{y} = (0.0, 0.0, 0.0, 0.0, 0.0)$ and transform the other 150 texts into texts with all positive sentiments $\boldsymbol{y} = (1.0, 1.0, 1.0, 1.0, 1.0)$. We evaluate the sentiment accuracy (Acc) of generated texts towards different aspects by a FastText classifier [13] trained on the training data. Moreover, we employ three workers on AMT to score each of them according to sentiment accuracy (Att), preservation of content (Con) and fluency (Gra), the same as before.

The results are shown in Table 4, and some cases are shown in **Supplementary Material**. We see that the achieved sentiment accuracy is high, which means that our model can perform sentiment transfer over multiple aspects at the same time. Considering the results of human evaluation, our model has good fluency and preservation of content when performing sentiment transferring over multiple aspects. To the best of our knowledge, this is the first work investigating the aspect-based attribute transfer task.

## 4.5 Transfer Degree Control

As is mentioned before, our model can use modification weight in $\boldsymbol{w}$ to control the degree of attribute transfer. Further, We want to have an insight into the impact of different $\boldsymbol{w}$ on the modification results.

Table 4: Results for multi-aspect attribute transfer. The kappa coefficient of the three workers is 0.67 $\in$ (0.61, 0.80), which means that the consistency is substantial.

| Aspects | Acc | Att | Con | Gra |
|---|---|---|---|---|
| Appearance | 90.2% | 3.2 | 3.5 | 3.8 |
| Aroma | 89.3% | 3.4 | 3.9 | 3.7 |
| Palate | 91.2% | 3.1 | 3.8 | 3.7 |
| Taste | 88.2% | 3.4 | 3.7 | 3.6 |
| Overall | 87.3% | 3.6 | 4.0 | 3.8 |

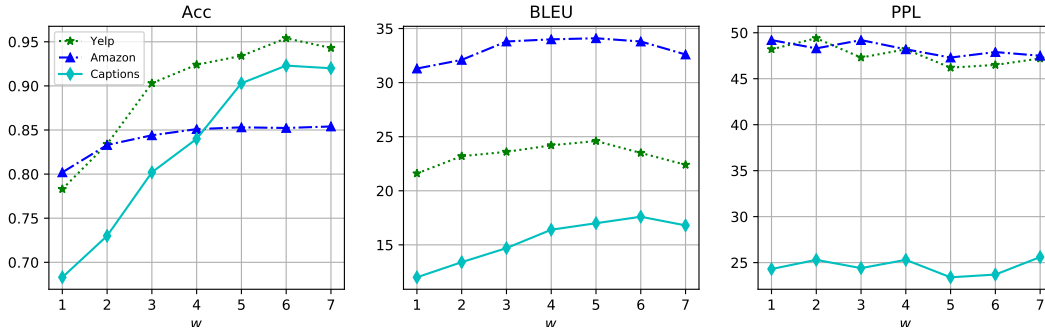

Figure 2: Influence of the modification weight $w$.

We let $w$ contain only one value and then let the value change from small to large, the visualization results are shown in Figure 2. Some conclusions can be concluded from the results: 1) As the value in $w$ increases, the attribute of the generated sentence becomes more and more accurate. 2) However, the BLUE score first increases and then decreases, we argue that this is because the attribute of some human-written references is not obvious. 3) PPL has not changed so much, which proves the effectiveness of our autoencoder with low reconstruction bias, and our latent representation editing method does not damage the fluency and naturalness of the sentence.

We also show two examples in Yelp test dataset in Table 5 (more cases are shown in **Supplementary Material**). From the table, we can see that as the value in $w$ increases, the target attribute of the generated sentence becomes more obvious. To the best of our knowledge, our model is the first one that can control the degree of attribute transfer.

### 4.6 Latent Representation Modification Study

In order to illustrate the latent representation editing result more clearly, we use T-SNE [21] to visualize the latent representation in the modification process. More specifically, we present latent representations of Yelp's test dataset (source), and the modified latent representations with different transfer degree weight in $w$, as shown in Figure 3.

Table 5: Examples of generation with different modification weight $w$.

| | Positive ->Negative | Negative ->Positive |
|---|---|---|
| Source: | really good service and food . | it is n't terrible , but it is n't very good either . |
| Human: | the service was bad | it is n't perfect , but it is very good . |
| $w = \{1\}$ | really good service and food . | it is n't terrible , but it is n't very good either . |
| $w = \{2\}$ | very good service and food . | it is n't terrible , but it is n't very good delicious either . |
| $w = \{3\}$ | very good food but service is terrible ! | it is n't terrible , but it is very good delicious either . |
| $w = \{4\}$ | not good food and service is terrible ! | it is n't terrible , but it is very good and delicious . |
| $w = \{5\}$ | bad service and food ! | it is n't terrible , but it is very good and delicious appetizer . |
| $w = \{6\}$ | very terrible service and food ! | it is excellent , and it is very good and delicious well . |

From Figure 3, we can see that the original representations of positive texts and negative texts are mixed together in the latent space. However, as the value in $w$ increases, the distinction between the modified latent representations of positive texts and negative texts becomes more and more obvious, which can also prove the effectiveness of using $w$ to control the degree of attribute transfer.

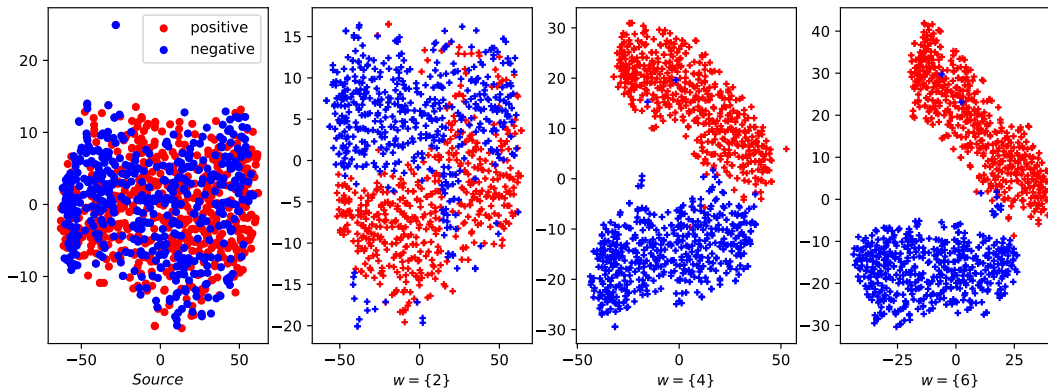

Figure 3: Visualization of representations with different modification weight $w$.

## 5    Conclusion and Discussion

In this work, we present a controllable unsupervised text attribute transfer framework, which can edit the entangled latent representation instead of modeling attribute and content separately. To the best of our knowledge, this is the first one that can not only control the degree of transfer freely but also perform sentiment transfer over multiple aspects at the same time. Nevertheless, we find that there may be some failure cases, such as learning some attribute-independent data bias or just adding phrases that match the target attribute but are useless (some cases are shown in Supplementary Material). Therefore we will try to further improve the performance in the future.

## Acknowledgments

This work was supported by National Natural Science Foundation of China (61772036) and Key Laboratory of Science, Technology and Standard in Press Industry (Key Laboratory of Intelligent Press Media Technology). We appreciate the anonymous reviewers for their helpful comments. Xiaojun Wan is the corresponding author.

## Footnotes

[1]Our codes are available at https://github.com/Nrgeup/controllable-text-attribute-transfer

[2]https://github.com/moses-smt/mosesdecoder/blob/master/scripts/generic/multi-bleu.perl

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
