[Supplementary Material]

# Supplementary Material

## 0.1 Style Transfer Cases

| Negative -> Positive (Yelp) | |
|---|---|
| Input: | it is n't terrible , but it is n't very good either . |
| CrossAlign: | it is always great , but it is always very good . |
| MultiDec: | it is n't delicious but , it is not very well good . |
| StyleEmb: | it is n't terrible , but it is n't very good either . |
| CycleRL: | it is n't recommend , but it is n't very great variations . |
| BackTrans: | she was very nice , but the service is great . |
| RuleBase: | it is n't oh so good ! but it is |
| DelRetrGen: | it is n't great but it is reasonably priced . |
| UnsupMT: | it is excellent , but it is good too . |
| Ours: | **it is excellent , and it is very good delicious .** |
| Human: | it is n't perfect , but it is very good . |
| Input: | the food was so-so and very over priced for what you get . |
| CrossAlign: | the food was fantastic and very very nice for what you . |
| MultiDec: | the food was low up and over great , see you need . |
| StyleEmb: | the food was so-so and very over priced for what you get . |
| CycleRL: | the food was so-so and very over priced for what great qualities . |
| BackTrans: | the food is delicious and the staff are very good for me . |
| RuleBase: | the food was so-so and very over priced for what you get just right . |
| DelRetrGen: | the service is fantastic and the food was so-so and the food is very priced for what you get . |
| UnsupMT: | the food was decent and very perfectly priced for what you get . |
| Ours: | **the food was great and very perfectly priced for what you get .** |
| Human: | the food was great and perfectly priced |
| Input: | i guess she was n't happy that we were asking the prices . |
| CrossAlign: | i guess she was n't great and we were pretty good . |
| MultiDec: | i would maybe worked that i been taking they were but very helpful . |
| StyleEmb: | i guess she was n't happy that we were n't more service . |
| CycleRL: | i guess she was n't happy that we were asking great qualities . |
| BackTrans: | i think i will be back to go here , we love it . |
| RuleBase: | i very reasonable and happy that we were asking the prices . |
| DelRetrGen: | fast , fast , happy that we were asking , and the prices were great . |
| UnsupMT: | i think she was happy that we are attentive the prices . |
| Ours: | **i guess she was happy that we were asking the prices .** |
| Human: | she was happy that we were asking for the prices . |
| **Positive -> Negative (Yelp)** | |
| Input: | the service was top notch and the food was a bit of heaven . |
| CrossAlign: | the service was top notch and the room was very expensive on me . |
| MultiDec: | the service was top and the food was a bit of this plate . |
| StyleEmb: | the service was top notch and the food was a bit of heaven . |
| CycleRL: | the service was top that awful and the dinner was not great existent . |
| BackTrans: | the service is horrible and the food is not very good . |
| RuleBase: | slow the food was a bit of |
| DelRetrGen: | the food was a bit of weird . |
| UnsupMT: | the service was lacking and the food was a bit of sick . |
| Ours: | **the service was bad and the food was a bit of sick .** |
| Human: | the service was bad and the food was hellish |
| Input: | food is always amazing no matter what i order . |
| CrossAlign: | food is does not sense about what i did n't care . |
| MultiDec: | food is always better no do so you have . |
| StyleEmb: | food is always amazing no matter what i order . |
| CycleRL: | food is always no amazing matter what i order . |
| BackTrans: | really awful and chocolate service . |
| RuleBase: | awful no matter what i order . |
| DelRetrGen: | no matter what i repeated order . |

| | |
|---|---|
| UnsupMT: | food is always horrible no clue what i order . |
| **Ours:** | **food is awful no matter what i order .** |
| Human: | the food is always terrible when i order here . |

| | |
|---|---|
| Input: | i 'll definitely go back here again . |
| CrossAlign: | i 'll not go back here again again again . |
| MultiDec: | i 'll go definitely back here again . |
| StyleEmb: | i 'll definitely go back here again . |
| CycleRL: | i 'll great judgement back again . |
| BackTrans: | i am i 've not . |
| RuleBase: | rude staff and terrible here again . |
| DelRetrGen: | i refuse to go here again . |
| UnsupMT: | i 'll probably not go back here again . |
| **Ours:** | **i 'll never go back here again .** |
| Human: | i wo n't go back there . |

**Negative -> Positive (Amazon)**

| | |
|---|---|
| Input: | this is not worth the money and the brand name is misleading . |
| CrossAlign: | this is not the best and the best is not great . |
| MultiDec: | this is not worth the money and this pan , at amazon . |
| StyleEmb: | this is not worth the money and the brand name is the price . |
| CycleRL: | this is not worth the money and the brand name minimizes rental . |
| BackTrans: | this is not a great biased , and the de de . |
| RuleBase: | you can not beat the price and the brand name is misleading . |
| DelRetrGen: | well worth the money and the brand name is misleading . |
| UnsupMT: | this is definitely worth the money and the brand name is illustrated . |
| **Ours:** | **this is worth the money and the brand name is great .** |
| Human: | this is worth the money and the brand name is awesome . |

| | |
|---|---|
| Input: | i could barely get through it they taste so nasty . |
| CrossAlign: | i ve had it for my husband and they are . |
| MultiDec: | i could do take it all them so easy so . |
| StyleEmb: | i could smell of smell of it right and helps . |
| CycleRL: | i could barely get through it they taste so nasty . |
| BackTrans: | i can t be happy with this product . |
| RuleBase: | beautifully through it they taste so nasty . |
| DelRetrGen: | i have used it through and it is very sharp and it was very nasty . |
| UnsupMT: | i can perfect get through it they taste so delicious . |
| **Ours:** | **i totally noticed they taste so good .** |
| Human: | i loved it because they taste so great . |

| | |
|---|---|
| Input: | i will never again purchase another game where this is a requirement . |
| CrossAlign: | i ve never had this for my wife and it s a num_extend |
| MultiDec: | i will never use these out of this , is the best . |
| StyleEmb: | i will never purchase again another game you ll put a bad . |
| CycleRL: | i will never again purchase another game where this is dangerously floured . |
| BackTrans: | i can t use it for my phone , but it s une . |
| RuleBase: | i will never again purchase measuring cup this is a requirement . |
| DelRetrGen: | i will never again purchase another one in the game where this is a requirement . |
| UnsupMT: | i will definitely again purchase another pan where this is a cannot . |
| **Ours:** | **i will purchase this game again where it is a requirement .** |
| Human: | i will purchase this type of game again . |

**Positive -> Negative (Amazon)**

| | |
|---|---|
| Input: | i would definitely recommend this for a cute case . |
| CrossAlign: | i would not recommend this for a long time . |
| MultiDec: | i would definitely recommend this for a bra does it . |
| StyleEmb: | i would definitely recommend this for a cute case . |
| CycleRL: | i wish of this product the promises online terrible . |
| BackTrans: | midnight i have to get this product for a un . |
| RuleBase: | skip this one for a cute case . |
| DelRetrGen: | i would not recommend this for a cute case . |
| UnsupMT: | i would definitely not recommend this for a cute case . |

| | |
|---|---|
| Ours: | **i would definitely not recommend this for a cute case .** |
| Human: | i would definitely not recommend this for a cute case . |

| | |
|---|---|
| Input: | very nice unit , easy to assemble and operate . |
| CrossAlign: | very good , but you are very happy with . |
| MultiDec: | very nice unit , easy to wear and tastes . |
| StyleEmb: | very good in case , it feels excellent results . |
| CycleRL: | very clip , did to [UNK] deliver and punch . |
| BackTrans: | no texture , and et t have to get the hair . |
| RuleBase: | very nice unit didn t work at assemble and operate . |
| DelRetrGen: | very nice unit and was looking forward to this operate . |
| UnsupMT: | very nice unit , impossible to assemble and operate . |
| Ours: | **very terrible unit , hard to assemble and boot .** |
| Human: | very ugly unit , hard to assemble and difficult to operate . |

| | |
|---|---|
| Input: | i wouldn t trade them but this is a great knife for the price . |
| CrossAlign: | i don t know it s just like the same for a few days . |
| MultiDec: | i can t return it and this product a great game for the price . |
| StyleEmb: | i wouldn t believe it to say the last such a decent use price . |
| CycleRL: | i [UNK] t trade them but this is a great glitch containing [UNK] smoking . |
| BackTrans: | i don t buy this product , but it s not a le . |
| RuleBase: | i i wish i hadn them but price . |
| DelRetrGen: | i wouldn t trade them or recommend this knife for the price . |
| UnsupMT: | i wouldn t trade them but this is a horrible toy for the price . |
| Ours: | **i would trade them but here it is a horrible knife for the price .** |
| Human: | i would trade them because this is a horrible knife for the price . |

**Factual -> Romantic (Captions)**

| | |
|---|---|
| Input: | a man and woman against a pink background smile . |
| CrossAlign: | a man in a red shirt is running on a beach . |
| MultiDec: | a man and woman on a red crowd looks . |
| StyleEmb: | a man and a woman smile and talk with one . |
| CycleRL: | a man and woman against a pink background smile . |
| BackTrans: | a man and woman against a pink background smile . |
| RuleBase: | a man and woman against a pink background smile loved . |
| DelRetrGen: | a man and woman watches a pink street to show his lover . |
| UnsupMT: | a man and woman crossing a kiss together dreaming of love . |
| Ours: | **a man and a woman hug with a pink smile on face .** |
| Human: | a bearded man and a woman in a dress holding a cup with a smiley face . |

**Factual -> Humorous (Captions)**

| | |
|---|---|
| Input: | a young man dances by a fountain . |
| CrossAlign: | a man is running on a beach to find the space . |
| MultiDec: | a young man stands next like a car . |
| StyleEmb: | a young man dances along an inflatable fountain . |
| CycleRL: | a man dances by a fountain . |
| BackTrans: | a young man dances by a fountain . |
| RuleBase: | a young man dances by a fountain deadly . |
| DelRetrGen: | a young man is running off for supremacy . |
| UnsupMT: | a young man sits by a fountain like a monkey with a smiley face . |
| Ours: | **a young man dances by a fountain in a yellow jacket , playfully playing .** |
| Human: | a boy in dark clothing near fountain water spout trying play with water . |

## 0.2 Transfer Degree Control Cases

**Negative -> Positive**

| | |
|---|---|
| Source: | it is n't terrible , but it is n't very good either . |
| Human: | it is n't perfect , but it is very good . |
| $w = 1.0$: | it is n't terrible , but it is n't very good either . |
| $w = 2.0$: | it is n't terrible , but it is n't very good delicious either . |
| $w = 3.0$: | it is n't terrible , but it is very good delicious either . |

| | |
|---|---|
| $w = 4.0$: | it is n't terrible , but it is very good and delicious . |
| $w = 5.0$: | it is n't terrible , but it is very good and delicious appetizer . |
| $w = 6.0$: | it is excellent , and it is very good and delicious well . |

| | |
|---|---|
| Source: | there is definitely not enough room in that part of the venue . |
| Human: | there is so much room in that part of the venue |
| $w = 1.0$: | there is definitely not enough room in that part of the venue . |
| $w = 2.0$: | there is definitely enough amazing room in that part of the venue . |
| $w = 3.0$: | there is definitely amazing enough room in that part of the venue . |
| $w = 4.0$: | there is definitely amazing enough room in that amazing part . |
| $w = 5.0$: | there is definitely amazing enough room in that well of the hidden gem . |
| $w = 6.0$: | there is definitely amazing nice room , and the part of amazing phenomenal ! |

| | |
|---|---|
| Source: | definitely disappointed that i could not use my birthday gift ! |
| Human: | definitely not disappointed that i could use my birthday gift ! |
| $w = 1.0$: | definitely disappointed that i could not use my birthday gift they . |
| $w = 2.0$: | definitely disappointed that i could use not got my birthday gift questions ! |
| $w = 3.0$: | definitely liked you and i could use my birthday gift easy that ! |
| $w = 4.0$: | definitely liked you and i 'll seen my birthday gift both a message ! |
| $w = 5.0$: | definitely recommend and i could enjoys all my birthday gift free salsa ! |
| $w = 6.0$: | definitely recommend and i 'll answered all my birthday gift from friend ! |

| | |
|---|---|
| Source: | we sit down and we got some really slow and lazy service . |
| Human: | the service was quick and responsive |
| $w = 1.0$: | we sit down and we got some really slow and lazy service . |
| $w = 2.0$: | we sit down and we got some really nice slow and lazy service . |
| $w = 3.0$: | we sit down and we got some really nice fast and decorated service . |
| $w = 4.0$: | we sit down and we got really some great quick and organized service . |
| $w = 5.0$: | we sit down and we got some really quick great fly and decent service . |
| $w = 6.0$: | we enjoy nice brunch and we got really quick friendly service and have great fun . |

| | |
|---|---|
| Source: | there chips are ok , but their salsa is really bland . |
| Human: | these chips are okay but their salsa is really tasty |
| $w = 1.0$: | there chips are ok , but their salsa is really bland and amazing . |
| $w = 2.0$: | there chips are ok , but their salsa is really nice and tasty . |
| $w = 3.0$: | there chips are ok , but their salsa is really nice tasty ! |
| $w = 4.0$: | there chips are ok , their salsa is amazing really enjoy ! |
| $w = 5.0$: | there are ok chips forward to their salsa and is amazing great ! |
| $w = 6.0$: | there are nice chips , their salsa is amazing great ! |

| | |
|---|---|
| Source: | the wine was very average and the food was even less . |
| Human: | the wine was above average and the food was even better |
| $w = 1.0$: | the wine was very average and the food was even less than needs . |
| $w = 2.0$: | the wine was very nice and the food was even less dining . |
| $w = 3.0$: | the wine was very nice and the food was easy even above business . |
| $w = 4.0$: | the wine was very amazing and the food was above happy . |
| $w = 5.0$: | the wine was very amazing and the food is large always fun . |
| $w = 6.0$: | great wine is very amazing and the food always reasonable . |

| | |
|---|---|
| Source: | the burgers were over cooked to the point the meat was crunchy . |
| Human: | the burgers were cooked perfectly and the meat was juicy |
| $w = 1.0$: | the burgers were over cooked to the point the meat was crunchy . |
| $w = 2.0$: | the burgers were over cooked to the point the meat was crunchy perfect . |
| $w = 3.0$: | the burgers were over cooked to the point the meat was crunchy perfect . |
| $w = 4.0$: | the burgers were fantastic from top to the the point meat was great crunchy . |
| $w = 5.0$: | the burgers were fantastic from always the point the service is perfect crunchy . |
| $w = 6.0$: | the burgers were fantastic from always the meat is perfectly . |

| | |
|---|---|
| Source: | said we could n't sit at the table if we were n't ordering dinner . |
| Human: | they said we could sit at the table with no hesitation |
| $w = 1.0$: | said we could n't sit at the table if we were n't ordering dinner . |
| $w = 2.0$: | said we could n't sit at the table if we were ordering a dinner ! |
| $w = 3.0$: | said we nice as the table of the brunch if we were n't ordering ! |
| $w = 4.0$: | said we lovely as the table for the brunch , we were n't ordering ! |

| | |
|---|---|
| $w = 5.0$: | said we lovely as the table for brunch from the dinner we were always ! |
| $w = 6.0$: | said we lovely as the table for brunch from the dinner we were always ! |

**Positive -> Negative**

| | |
|---|---|
| Source: | really good service and food . |
| Human: | the service was bad |
| $w = 1.0$: | really good service and food . |
| $w = 2.0$: | very good service and food . |
| $w = 3.0$: | very good food but service is terrible ! |
| $w = 4.0$: | not good food and service is terrible ! |
| $w = 5.0$: | bad service and food ! |
| $w = 6.0$: | very terrible service and food ! |

| | |
|---|---|
| Source: | they were so helpful , kind , and reasonably priced . |
| Human: | they should 've been more helpful , kind , and reasonably priced . |
| $w = 1.0$: | they were so helpful , kind , and reasonably priced , slow . |
| $w = 2.0$: | they were so helpful , kind , and priced were bad . |
| $w = 3.0$: | they were so helpful , kind , and noticed priced no reasonably . |
| $w = 4.0$: | they were so only rude , although , one were sort about fries . |
| $w = 5.0$: | they were so only rude , although , took not minutes they nasty . |
| $w = 6.0$: | they were so only rude , nothing but bad was noticed . |

| | |
|---|---|
| Source: | the atmosphere was fun and the staff treats you well . |
| Human: | the atmosphere was lame and the staff treats you like dirt |
| $w = 1.0$: | the atmosphere was fun and the staff shut you totally said the well . |
| $w = 2.0$: | the atmosphere was not fun and the staff said they totally let it well . |
| $w = 3.0$: | the atmosphere was not fun and the staff said it was wrong . |
| $w = 4.0$: | the atmosphere was not fun and the toilet was bad . |
| $w = 5.0$: | the atmosphere was not that but the staff was bad and wrong . |
| $w = 6.0$: | the atmosphere was trash and the staff bad ! |

| | |
|---|---|
| Source: | i also love their convenient location right off of scottsdale road . |
| Human: | i hate how their location is remote to get to from scottsdale road |
| $w = 1.0$: | i also love their convenient location right off of scottsdale road ! |
| $w = 2.0$: | i also not love their convenient location right off of scottsdale road . |
| $w = 3.0$: | i can not drive to their location though right off of scottsdale road . |
| $w = 4.0$: | i might not drive to their location right off of scottsdale road . |
| $w = 5.0$: | i apparently not driving off their purchase location, it was n't short management . |
| $w = 6.0$: | i hate driving to their location right off of scottsdale road . |

| | |
|---|---|
| Source: | i actually can not wait to come back next year ! |
| Human: | i would not return here next year |
| $w = 1.0$: | i actually can wait to come back next year ! |
| $w = 2.0$: | i actually not come back next year . |
| $w = 3.0$: | i actually could n't wait to come back this bad place next year ? |
| $w = 4.0$: | i actually could n't wait to said wrong service . |
| $w = 5.0$: | i could n't wait to said this wrong service two minutes ? |
| $w = 6.0$: | i could n't wait to said this wrong service was no tables . |

| | |
|---|---|
| Source: | everything is fresh and so delicious ! |
| Human: | everything was so stale |
| $w = 1.0$: | everything is fresh and so salty , but three spot is delicious ! |
| $w = 2.0$: | everything is fresh and so salty , but absolutely _num_ smell delicious ! |
| $w = 3.0$: | everything is no and later but only smelled nasty , it was cold ! |
| $w = 4.0$: | service is half but to only later but his smelled bad ! |
| $w = 5.0$: | everything is terrible and no closed toilet |
| $w = 6.0$: | everything is terrible and smelled nasty ! |

| | |
|---|---|
| Source: | great food recommendations steak and tuna were both great . |
| Human: | the steak and tuna were not up to par |
| $w = 1.0$: | great food recommendations and steak tuna were both great the lost food . |
| $w = 2.0$: | food ordered recommendations steak and tuna were both terrible but got more . |
| $w = 3.0$: | food ordered steak recommendations but tuna were finally flat had bad wrong . |
| $w = 4.0$: | ordered food steak but were tuna was both ok my bad mine . |

| | |
|---|---|
| $w = 5.0$: | ordered food were steak but was already tasted disappointed the food . |
| $w = 6.0$: | but noticed mine were bad and i already left flat poor food . |

| | |
|---|---|
| Source: | i was nervous and she made me feel so comfortable and welcome . |
| Human: | she did not make me feel welcomed |
| $w = 1.0$: | i was nervous and she made me feel so comfortable and welcome . |
| $w = 2.0$: | i was nervous and she made me feel so comfortable and mad . |
| $w = 3.0$: | i was nervous and she made me feel so busy . |
| $w = 4.0$: | i was nervous and she is so busy and made me feel mad . |
| $w = 5.0$: | i was nervous and she made me feel wrong and so mad . |
| $w = 6.0$: | i was nervous and she made me sad and felt mad . |

## 0.3 Multi-aspect Attribute Transfer Cases

| Source:[0.9, 1.0, 0.9, 0.8, 0.9] | Target:[0.0, 0.0, 0.0, 0.0, 0.0] |
|---|---|
| from a can, pours straw gold with thick white head ; aroma is pine and sugar , some citrus but mostly pine . taste is fruity , citrus , tea , slight pine , toast grain. mouthfeel is spot on, carbonation is medium , this brew is quite refreshing ! definitely something i wanna buy in quantity to go camping with this summer ! | from a can, pours yellow cloudy lead ; aroma is spicy grassy hop, some citrus but mostly pine . taste is fruity , citrus , tea , slight pine , toast grain. mouthfeel is spot on, carbonation is medium , and unimpressive flavor character ! definitely i will never buy in quantity to go camping with this summer ! |
| Source:[0.5, 0.3, 0.7, 0.3, 0.3] | Target:[1.0, 1.0, 1.0, 1.0, 1.0] |
| this beer pours a pale straw color . the head is a half an inch in height , and recedes quickly leaving no lacing . the aroma is of nothing. when i try really hard i can detect some extremely faint malt . the taste is little easier to detect that the aroma, but it is still mostly nothing . the mouthfeel is light bodied with nice carbonation . it is actually pretty good for a light lager. overall, this beer tastes too much like water for my likings . i will not be having this beer again . | this beer pours a deep red, glowing, standing out a lot , and recedes quickly . the aroma is is of smoked meat , smoked cheese. the taste is little easier to detect that the aroma, but it is still very good . the mouthfeel is light bodied with nice carbonation . it is actually pretty good . overall, this beer tastes like my likings . i will try this beer again . |

## 0.4 Transformer-based Auto-encoder Reconstruction Low Bias

| | |
|---|---|
| Encode: | ever since joes has changed hands it 's just gotten worse and worse . |
| Decode: | ever since joes has changed hands it 's just gotten worse and worse . |
| Encode: | there is definitely not enough room in that part of the venue . |
| Decode: | there is definitely not enough room in that part of the venue . |
| Encode: | she said she 'd be back and disappeared for a few minutes . |
| Decode: | she said she 'd be back and disappeared for a few minutes . |
| Encode: | definitely disappointed that i could not use my birthday gift ! |
| Decode: | definitely disappointed that i could not use my birthday gift ! |
| Encode: | the atmosphere was fun and the staff treats you well . |
| Decode: | the atmosphere was fun and the staff treats you well . |
| Encode: | their pizza is the best i have ever had as well as their ranch ! |
| Decode: | their pizza is the best i have ever had as well as their ranch ! |
| Encode: | i also love their convenient location right off of scottsdale road . |
| Decode: | i also love their convenient location right off of scottsdale road . |

## 0.5 Failure Cases

| |
|---|
| Adding meaningless phrases |

| | |
|---|---|
| Source: | just left and took it off the bill . (Negative) |
| Output: | just left and took it off the bill . great fun . (Positive) |

**Attribute-independent data bias**

| | |
|---|---|
| Source: | everything is fresh and so delicious ! |
| Output: | everything is terrible and no closed toilet |