[Reviews · NeurIPS 2019]

Reviewer 1



Detailed comments on the contributions: Contribution 1: The learning model consists of an autoencoder that is based on the Transformer. The latent variable z of the autoencoder is supposed to be fed to a classifier that is trained separately and that will guide the autoencoding process. Questions: If the classifier C is used to change z and so its parameters are kept fixed, how is it insured that this classifier has the optimal set of parameters as z is changed? The idea of “control of degree of transfer” seems a bit not well-grounded to me… Language is complex, and I find it hard to draw a parallel between fine-tuning one weighting parameter in an objective function and an effect on the output where, as I understand what you mean by it, certain words will change in, e.g., intensity? [e.g. funny -> hilarious]… Contribution 2: By way of its design, the model allows (at least conceptually) to handle multiple attributes effectively for transfer. Experiment/Section 4.2: The authors choose the most straightforward experiment. Given how strongly the authors claim their model to be performing in previous sections, it would’ve made sense to evaluate the model on cases that are bit more involved. Another challenging dataset is the one introduced by Ficler and Goldberg 2017 (full title later in the review) in which they do multi-field conditioned generation. That dataset has 5 fields that the author could use to further investigate their model’s performance. Contribution 3: The authors perform extensive experiments against 8 previous models. The experiment seem quite comprehensive in terms of competing models. Given the strong performance of the model (surprisingly in almost all of their test cases as in Table 2), the authors could've chosen some more involved qualitative validation in Section 4.2 as noted above. I'm also hesitant regarding some of the conclusions regarding transfer control. In my opinion, the contribution of these results (e.g. Figure 2) is the performance trends they show the model to exhibit when the parameter is tuned, but this is not a general indication of the impression that the paper is trying to give regarding the effect of fine-tuning that weight parameter [the actual control of word usage]. ------------------------------------------------------------------------------------ Clarity: The paper reads very smoothly and is an enjoyable read. The model is explained clearly and the paper is well-structured. The description of Algorithm 1 is useful. However, I find it crucial that the author also explain how the two components of their model interact with each other. This is missing and I think it's crucial. If the paper is accepted, given the extra space, I encourage the authors to further explain that [other than the fact that they're trained separately]. Intro: What does this mean? “this may undermine the integrity” Intro: At this point of the paper, it's unclear what the notion “degree of transfer” means. It should be clarified in the intro. Table 2: The way numbers are bolded vs underlined is confusing. Bold is usually used to highlight the strongest performance, whereas the authors use the underlining to point to strongest performance. I encourage the authors to bold the strongest numbers for the Captions dataset for the corresponding (competing) models and keep the underlining to indicate that this is their numbers. If the paper is accepted, given the extra space, I encourage the authors to explain better their "Automatic Evaluation" results of Section 4.1 drawing parallels between the trends in numbers and the way they are explained in the body of the section. ------------------------------------------------------------------------------------ Additional references: Since the authors mention at the beginning of their “Related Work” section the notion of transfer of style as a whole (not attributes) [specifically in Lines 61-62], these are possible references that could be added for that: 2012: Paraphrasing for Style 2016: Stylistic Transfer in Natural Language Generation Systems Using Recurrent Neural Networks 2017: Shakespearizing Modern Language Using Copy-Enriched Sequence-to-Sequence Models 2018: Evaluating prose style transfer with the Bible Not transfer per se but relevant to generation of texts conditioned on specific attributes (the paper I mentioned above): 2017: Controlling Linguistic Style Aspects in Neural Language Generation ------------------------------------------------------------------------------------ Typos and other comments: Line 51: fluent* Line 55: capable of* controlling* Line 85: edit* Line 116: language modelling* Line 129: “representation that conforming to the target” -> “representation so that it conforms to the target” Line 136: editing the* latent representation Line 141: “z’ that can be identified as the target attribute y’ ”. I understand this to mean that z’ is the target attribute y’… I think what you’re trying to say “ z’ that leads to the identification of target attribute y’ “? Order of papers in References is off at some point (e.g., Smola et al appearing at [1], Van Der Maarten at al. appearing at [18] and not towards the end…)

Reviewer 2



The paper presents an approach to the increasingly popular problem of "Text Attribute Transfer/Text Style Transfer". The authors motivate their approach by pointing to prior work that argues that disentanglement of "content" and "attribute" in the latent representation of a sentence isn't a necessity to achieve attribute transfer. They therefore present an approach that seeks to "edit" an entangled latent representation, via gradient signals from a pre-trained, differentiable, classifier trained on the same latent representations, to alter the sentence's attribute. Entangled latent representations are produced by a simple sequence autoencoder. Pros: - Well rounded experiments and evaluation (3 automatic metrics that account for different aspects of the problem - BLEU, PPL, classifier accuracy + human evaluation) on 4 datasets. - Good improvements on multiple metrics including human evaluation on multiple benchmarks. - Although I don't find much discussion about this, the approach presented in this paper should be a lot faster to train that unsupervised MT like approaches that rely on backtranslation. The autoencoder and classifiers can be pre-trained independently and the activation maximization step needs to be run only at inference. - Interesting overall approach, investigating the broad viability of latent space editing via activation maximization in NLP and specifically for text style transfer. Concerns: - Gains over prior work seem quite substantial, but as a baseline, I would have liked to have seen a simpler autoencoder architecture. It isn't clear to me how much of the gains come from this particular choice of architecture. - When presenting how well the autoencoder performs at reconstructing sentences, qualitative examples, seem like a bad choice to demonstrate this behaviour (Table 0.4 in the supplementary section), presenting BLEU scores between input and reconstructions should paint a better picture. - Related work and presentation of the paper make connections to the FGSM adversarial attack, but not to a large body of work on activation maximization, that are more related to this line of work, since they've been successfully applied to producing generative models (see for example [1] Plug & Play Generative Networks: Conditional Iterative Generation of Images in Latent Space by Nguyen et al. (2017) [2] Synthesizing the preferred inputs for neurons in neural networks via deep generator networks by Ngyuyen et al. (2016) [3] Visualizing higher-layer features of a deep network by Erhan et al. (2009).) Having read the authors' response, I still think this is a good submission and look forward to seeing their ablation studies.

Reviewer 3



Originality: The methods mentioned in the paper are a novel combination of well-known prior techniques. The method is a combination of the work in Lample et al, 2019 along with the change that this work does not require an embedding for the transfer attribute and uses the Fast-Gradient-Iterative Modification algorithm to gain a fine grained control over the degree of attribute transfer. It is clearly stated in the paper how it differs from prior contributions and the work is well cited. The tasks used in the paper are well defined in the prior work. Quality: The submission is well supported by experimental evidence and compares itself with a range of prior work. This is a complete piece of work and the authors have carefully evaluated the working of their approach except for the few point mentioned in the next section which can strengthen the work. Clarity: The paper is mostly clearly written and has provided clear implementation instructions to reproduce results except for the human evaluation section. It is unclear whether only 3 workers were hired to annotate all the 100-200 samples or 3 workers were hired for each of the test samples and then the majority vote as considered (Line 200). It is also not clear whether the references written by human for BLEU calculation are the original sentences before attribute transfer or there was some test data collected for acquire human references for attribute transfer (Line 186). Significance: In automated evaluation metrics the results are not as significant as they are in the human evaluation. Hence it is important to understand how human evaluation was performed. Did the same person annotate the same task for all the models? In general it is preferred to use A/B testing for human evaluations as the scale at which each person scores the sentences needs to be normalized. This is also backed by prior work on style transfer which uses this method of evaluation. The significant contribution of this paper comes from the novel approach of demonstrating how we can gain finer control over attributes and how changing the degree of attribute transfer can affect the generations.

[Author Response · NeurIPS 2019]

**Dear Reviewers,**

Thanks a lot for your helpful and insightful comments. We read them carefully and tried our best to address the issues raised in the comments. Below are our responses to the key issues in the reviews:

**To Reviewer #1:**

**1.** (For the problem that "How to ensure the classifier has the optimal set of parameters as z is changed") The classifier's parameters are kept fixed as z is changed. We train the autoencoder and the classifier on the training set, which is diverse and contains texts of varying degrees of attributes, reflected by the different confidence values given by the classifier. Thus we believe when the text in the test set is encoded into the same latent space as the training set, the classifier can continue to provide the optimal direction of modification, even when z changes.

**2.** (For the problem that "It's unclear what the notion 'degree of transfer' means") The 'degree of transfer' here means the apparent extent of the desired attributes embodied in the transfer results. Taking sentiment transfer as example, it is reflected in the sentimental intensity felt from the text (e.g., weak positive, moderate positive and strong positive). Different from most previous work that only provides binary control over attributes, one advantage of our model is the ability to give control over the degree of attribute transfer desired. Following the evaluation settings of previous work, we use various automatic evaluation indicators (ie, Acc, BLEU, and PPL in Section 4.1) to automatically evaluate transfer results from different aspects. Particularly, 'Acc' is used to evaluate the attribute's accuracy. In Figure 2, by showing the values of three automatic indicators under different fixed modification weights, we demonstrate the influence of the weight on controlling the transfer results. Sorry for the misunderstanding, we will add more words to explain this.

**3.** Thanks for your suggestion, we will add details about the interaction between the autoencoder and the classifier into Algorithm 1.

**4.** (For the problem that "What does 'this may undermine the integrity' mean ?") The word "integrity" here means naturality. For example, some phrase-based methods directly delete/replace/insert some sentimental words/phrases, which may result in the generated sentences that are unnatural and not as likely to be written by the human (can be partly reflected by the automatic indicator 'PPL'). Sorry for the misunderstanding, we will add more explanations.

**5.** Thank you for your careful review, we will further verify our model on more involved Multi-Aspect Sentiment Transfer dataset and multi-field conditioned generation dataset by Ficler and Goldberg 2017. We will improve our paper by adding more explanations about "degree of transfer" and "Automatic Evaluation" results, adding more comprehensive references, and carefully proofreading the paper.

**To Reviewer #2:**

**1.** (For concerns #1 and #2) We will compare our model with some simpler autoencoder architectures (e.g., LSTM, GRU), and show the BLEU scores between input and reconstructions. We will add more ablation&comparison experiments, to show the gains of different parts of our model.

**2.** (For concerns #3) Thanks for your insightful suggestions, we will add references about activation maximization including PPGN, DeVise, etc., and provide a more comprehensive related work section.

**To Reviewer #3:**

1. (For the problem that "how human evaluation was performed") For each test sample, we hired 3 workers to annotate and average the scores. Moreover, we divided the test samples of each model on each task into the same number of small sets, and the same person annotated the same task for all the models. Due a large number of samples to be labeled, we did not perform an A/B test, and we will add it if necessary. There was some test data collected for acquire human references for attribute transfer, and the references written by human for BLEU calculation are provided by previous work (Juncen Li, Robin Jia, He He, and Percy Liang. Delete, retrieve, generate: a simple approach to sentiment and style transfer.). Thanks for your suggestions, we will provide a clearer explanation to explain this.

2. (For the problem that "The argument about the correlation between human evaluation and automatic evaluation") The results of human evaluation and automatic evaluation are not always consistent with each other. In this study, we follow the evaluation settings of previous works to provide comprehensive results with various indicators. Thank you for your careful review, we will remove this argument about the correlation but provide more detailed evidence and analysis.

[Meta-Review · NeurIPS 2019]

All reviewers agree that this paper provides an original take on multi-attribute style transfer, with extensive experimentation. We believe this submission will be interesting to the NeurIPS audience and inspire fruitful research directions.